# Glycemic Variability and CNS Inflammation: Reviewing the Connection

**DOI:** 10.3390/nu12123906

**Published:** 2020-12-21

**Authors:** Charles Watt, Elizabeth Sanchez-Rangel, Janice Jin Hwang

**Affiliations:** Section of Endocrinology, Yale School of Medicine, New Haven, CT 06510, USA; charles.watt@yale.edu (C.W.); elizabeth.sanchezrangel@yale.edu (E.S.-R.)

**Keywords:** glycemic variability, oxidative stress, neuroinflammation, cognitive dysfunction, vascular dysfunction, endothelial inflammation, blood–brain barrier, diabetes

## Abstract

Glucose is the primary energy source for the brain, and exposure to both high and low levels of glucose has been associated with numerous adverse central nervous system (CNS) outcomes. While a large body of work has highlighted the impact of hyperglycemia on peripheral and central measures of oxidative stress, cognitive deficits, and vascular complications in Type 1 and Type 2 diabetes, there is growing evidence that glycemic variability significantly drives increased oxidative stress, leading to neuroinflammation and cognitive dysfunction. In this review, the latest data on the impact of glycemic variability on brain function and neuroinflammation will be presented. Because high levels of oxidative stress have been linked to dysfunction of the blood–brain barrier (BBB), special emphasis will be placed on studies investigating the impact of glycemic variability on endothelial and vascular inflammation. The latest clinical and preclinical/in vitro data will be reviewed, and clinical/therapeutic implications will be discussed.

## 1. Introduction

Glucose, the predominant energy source for the body and particularly for the brain [1], is transported across the blood–brain barrier and ultimately metabolized in the mitochondria to generate ATP [2]. In the process of oxidative phosphorylation, reactive oxygen species (ROS) are generated; thus, in disease states such as Type 1 and Type 2 diabetes, exposure to abnormal glucose levels can lead to high levels of oxidative stress and inflammation [3,4]. While much attention has been focused on the impact that chronic hyperglycemia [5] and hypoglycemia [6] have on the generation of oxidative stress and inflammation, a growing body of evidence has shown that fluctuations in blood glucose levels, or glycemic variability (GV), may also drive excessive ROS and oxidative stress and lead to vascular and cardiovascular complications [7,8,9]. This review will summarize the recent literature on the impact of fluctuations in glucose, neuroinflammation, and neurological function.

The definitions of glycemic variability have changed over time, particularly with the recent widespread use of technologies such as continuous glucose monitors (CGM), which were first approved in 1999 for clinical use in patients with Type 1 diabetes (T1DM) [10] and which measure interstitial blood glucose levels. Prior to CGM use, the term glycemic variability was often used to describe the variability between glycated hemoglobin (HbA1C) measurements or the direct blood glucose measurements made at clinical visits. These measurements were often taken weeks or months apart. With advances in CGM technology, the term “glycemic variability” has been adopted to indicate changes in peripheral glucose on the order of minutes rather than weeks. For coherence in this review, GV measured by inter-visit laboratory tests will be labeled “long-term” GV and GV measured through CGMs will be termed “short-term” GV. A short guide to common acronyms in GV is included in Table 1. For a more thorough review of GV metrics, see Rodbard 2009 [11] and Umpierrez 2018 [12].

It is important to highlight that in many studies, it may be difficult to distinguish between the effects of glycemic control and glycemic variability. Glycemic control is most commonly measured by the HbA1C (glycated hemoglobin fraction), which is the gold standard measurement for measuring the risk of developing diabetes-related complications [13,14]. The HbA1C is largely the result of a slow glycation of hemoglobin, which is dependent on circulating glucose levels and thus, is used as an average blood glucose concentration over the past ~3 months (the approximate life span of a typical red blood cell) [15]. Thus, the HbA1C does not provide any information about fluctuations in glucose levels during that period. Some clinical trials have reported that improved glycemic control can be associated with reduced glycemic variability [16]. Other studies have shown that the intensification of diabetes therapy leads to higher rates of hypoglycemia and variability [17]. Thus, it may be challenging to resolve the impact of glycemic control and variability independently. With the more frequent use of CGM technology as well as other biological markers such as 1,5 anhydro-d-glucitol [18] and glycated albumin [19], which may be more reflective of some measures of glycemic variation, such as postprandial glucose excursions, future studies may provide more insight into the independent effects of glucose control and variability.

## 2. Glycemic Variability, Oxidative Stress, and Inflammation

Reactive oxygen species are generated under normoglycemic conditions during cellular glucose metabolism [2,3]. While once considered simply by-products of cellular metabolism, ROS are now understood to play important roles in intracellular signaling, particularly in immune cells [20]. However, the balance between ROS production and antioxidant defense mechanisms is altered in states of hyper- and hypoglycemia. This imbalance then leads to increased oxidative stress as well as abnormal immune function and inflammation [4,21].

Hyperglycemia can drive excess ROS via multiple different pathways including: increased flux through the polyol pathway [22,23] and hexosamine pathway [24,25]; increased formation of advanced glycation end products [26,27]; and increased activation of protein kinase C through diacylglycerol [28,29]. A more detailed exploration of these mechanisms can be found in Brownlee 2001 [30] and it is very well-established that these underlying mechanisms contribute to the hyperglycemia-associated microvascular complications of diabetes including retinopathy, nephropathy, and neuropathy [30]. Although the mechanisms remain less well defined than in hyperglycemia, hypoglycemia also increases oxidative stress, likely via dysfunctional mitochondrial bioenergetics [31,32,33]. Mild hypoglycemia (2.5 mM) induces apoptosis and oxidative stress in cultured Schwann cells, and, in animal models, exposure to repeated episodes of hypoglycemia changes the expression levels of redox genes as well as increases levels of lipid peroxidation and protein carbonylation [34]. In patients with Type 2 diabetes mellitus (T2DM) studied using a hyperinsulinemic hypoglycemic clamp, mild hypoglycemia (2.7 mM) induces the production of markers of oxidative stress and inflammation including c-reactive protein and urinary free 8-isoprostoglandin F2α (8-iso PGF2α) [35].

Given the body of evidence that both hyperglycemia and hypoglycemia can independently lead to increased oxidative stress and inflammation, it is not surprising that fluctuations in glucose level may lead to even greater exposure to oxidative stress and inflammation [36,37]. The Diabetes Complication and Control Trial (DCCT) was the landmark trial that established a clear link between glycemic control (measured by HbA1C) and the risk of developing microvascular complications of diabetes [38]. However, while the DCCT found that mean HbA1C values were the dominant factor in predicting future microvascular complications, glycemic control did not solely account for the risk of complications [38], raising the possibility that additional factors such as glycemic variability could be contributing to patient outcomes [39,40]. To begin measuring the respective impact of hyperglycemia compared to glycemic variability, Monnier and colleagues [36] examined the relationship between CGM-measured metrics of glycemic variability and markers of oxidative stress. While individuals with T2DM had higher levels of 8-iso PGF2α compared to non-diabetic control subjects, there was no relationship between 8-iso PGF2α and any metrics of glycemic control including HbA1C or fasting glucose levels. However, higher glycemic variability was strongly correlated with higher levels of 8-iso PGF2α [36]. Two subsequent studies amongst individuals with T1DM [41] and T2DM [42] were not able to replicate these findings; however, there were differences in methodology as well as severity of diabetes in the subjects. More recently, plasma levels of 1,5-anhydroglucitol (1,5-AG) and glycated albumin, which may be more reflective of postprandial glucose excursions, [18] were found to be more closely associated with markers of oxidative stress than HbA1C [43]. Moreover, Ohara and colleagues reported improvements in markers of oxidative stress amongst 67 patients with T2DM following a 6-month intervention to reduce postprandial glucose excursions and glycemic variability [44]. While these studies point to a growing recognition that glycemic variability may be associated with oxidative stress and inflammation, they also highlight how differences in the methodology used to measure GV may have a significant impact on the interpretation of results. In fact, two separate reviews by Nalysnyk et al. [45] and Siegelaar et al. [46], which mainly focused on the impact of GV on peripheral complications of diabetes, drew fundamentally different conclusions about whether GV should be considered a risk factor for diabetes-related complications. With the recent publication of a consensus statement on the use of CGM to measure GV [47], future studies will include more standardized metrics for measuring GV, which will greatly improve the reproducibility of findings as well as improve our ability to interpret results.

Relatively fewer studies have focused specifically on the impact of GV on the central nervous system (CNS), despite strong evidence that both hyperglycemia and hypoglycemia have a profound impact on CNS function [48,49]. Thus, the remainder of this review will focus specifically on the impact of GV on the CNS (Figure 1).

## 3. Impact of GV on Central Nervous System Inflammation

The interactions between various brain cell types and the brain microvessel endothelial cells, often termed the neurovascular unit (NVU), are critical to maintaining CNS function. A growing body of the literature has described the impact of diabetes on components of the NVU, including: astrocytes, which are less populous and disconnected from endothelial cells [50]; pericytes, which are similarly diminished and retracted [51]; oligodendrocytes, which are decreased in number and short-lived [52]; microglia, which are abnormally polarized [53]; and endothelial cells, which are rendered less dense and more permeable [54,55,56]. However, very little is known about the impact of glycemic variability on the integrity of the NVU.

### 3.1. GV on Endothelium and Blood–Brain Barrier

Few studies have looked specifically at the impact of GV on brain endothelial function. In studies using brain endothelial cell lines, a brief acute exposure to hyperglycemia was associated with greater endothelial barrier dysfunction as measured by disruptions in transendothelial electrical resistance [57]. Similarly, in rodent models of diabetes, ex vivo analysis of brain microvessels revealed upregulation of inflammatory markers and evidence that glycemic variation leads to endothelial disruption [58]. These studies are consistent with other studies using human aortic endothelial [59] and umbilical [60,61] cell lines. In rodent models, glucose fluctuations have been associated with blood–brain barrier (BBB) dysfunction [62] as well as altered brain glucose transport [63]. Glut1 expression in brain endothelial cells has been shown in vitro to be increased following exposure to hypoglycemia [64] and the distribution of Glut1 on luminal compared to abluminal surfaces of microvessel endothelial cells may also be altered with hypoglycemia [63]. Conversely, chronic hyperglycemia may also be associated with decreased expression of Glut1 at the BBB [65]. One of the few studies that studied GV directly found that repeated glucose fluctuations had a larger effect on BBB transporters than acute, sustained glucose changes [58]. In humans, poorly controlled T1DM and T2DM have both been associated with diminished brain glucose [66,67].

In the human brain, virtually no studies have examined the impact of glycemic variability directly on the blood–brain barrier. Amongst T1DM patients, higher glycemic variability is associated with altered brain glucose transport capacity [68], which could have implications for hyperglycemia-driven ROS production and oxidative stress. A series of studies by Ceriello and colleagues used experimentally induced glucose fluctuations to investigate the impact of GV on endothelial function and inflammation. In one study, acute hyperglycemia resulted in peripheral endothelial dysfunction measured by flow-mediated dilation, increased inflammation measured by 8-iso prostaglandin F2α, and higher levels of oxidative stress measured by plasma nitrotyrosine [69]. Additional studies using similar experimental methods found that the degree of endothelial dysfunction and the levels of oxidative stress were higher with repeated glucose oscillations compared to a single step in glycemia [70]. Moreover, inducing extremes in glucose levels such as a period of hypoglycemia followed by hyperglycemia was associated with worse endothelial function as well as greater oxidative stress and inflammation [71]. More recently, amongst individuals with poorly controlled T2DM, higher CGM measured short-term GV, measured by mean absolute glycemic excursion (MAGE), was associated with an altered endothelial cell epigenetic profile on P66shc, an adapter protein that is a key driver of mitochondrial oxidative stress [72]. A similar study in patients with T1DM demonstrated a positive correlation between MAGE and higher levels of endothelial progenitor cells, which are typically produced to repair vascular damage [73].

### 3.2. GV and Microglia, Neuronal, and Astroglial Cells

Microglia, the resident macrophages of the central nervous system, comprise nearly 15% of cells in the brain [74] and play a critical role in synaptic pruning [75,76,77] and the phagocytosis of cellular debris [78]. Resting microglia are tightly regulated by interactions with neurons with microglia serving protective functions [79]. For example, when provided with signals that indicate the presence of tissue damage or pathogens, microglia become activated and carry out repair functions. However, excessive activation may lead to the release of inflammatory cytokines, chemokines, reactive oxygen species, and nitric oxide, which can lead to neuronal dysfunction and death [80,81,82,83]. Exposure to chronic hyperglycemia has been associated with activation of microglia [84]. Furthermore, when studying the mouse microglial BV-2 cell line, exposure to glucose fluctuations resulted in increased markers of metabolic stress leading to apoptosis and/or autophagy [85] as well as leading to shifts in microglial polarization to the inflammatory M1 phenotype [53].

Several studies have also shown the direct impact of GV on neuronal and astroglial cells [86,87]. An in vitro study of neuroblastoma cells found that 6-h fluctuations of glucose from 90 to 900 mg/dl resulted in decreased metabolic activity measured by a reduction in tetrazolium salts, and increased apoptotic gene expression [86]. A similar in vitro experiment on C6 astroglial cells exposed to hyperglycemia (12 mM) or glucodeprivation (0 mM) exhibited decreased cellular proliferation and glucose uptake as well as increased mitochondrial dysfunction, DNA damage, and ROS production [87]. An in vivo study comparing glucose fluctuation to constant hyperglycemia in female Goto-Kakizaki (GK) rats used twice daily intraperitoneal insulin injections to model GV and found that GV caused significantly more neuron apoptosis than hyperglycemia alone [88]. In addition, GV–GK rats had higher levels of inflammatory markers including tumor necrosis factor-alpha (TNFα) and interleukin-1beta [88]. A comparable study of male Sprague Dawley rats with streptozotocin-induced diabetes found similar results [89]. GV caused by the tri-daily alternation of intraperitoneal injections of glucose and insulin induced an inflammatory response measured by TNFα and interleukin-6 (Il-6) [89]. Furthermore, GV led to observable neuronal structural damage in myelin sheaths and axons when viewed via electron microscopy, which may explain the rats’ impaired performance on maze and passive avoidance tests [89].

### 3.3. GV and the Human Brain

Whether the mechanisms identified in animal models associating glycemic variability with neuroinflammation translate into the human brain is not clear and few studies have directly investigated this question. However, there is evidence that glycemic variability may be associated with impairments in cognitive function as well as decreased recovery from CNS injury.

Amongst patients with T2DM, in studies measuring long-term GV most [90,91,92], but not all [93], studies have found that greater long-term GV is associated with cognitive deficits. One large-scale study of over 11,000 older patients observed no relationships between HbA1C, variability in HbA1C, c-reactive protein levels, and cognitive performance [93]; however, several other studies amongst healthy, older patients with T2DM and no history of cognitive dysfunction found that long-term GV was associated with lower metrics of cognitive function [90,91] as well as with decreased levels of limbic and temporal–parietal gray matter [92]. Moreover, in a large cohort of >16,000 older patients with T2DM followed prospectively, long-term variation in HbA1C and variability in fasting plasma glucose levels was associated with an increased risk of developing Alzheimer’s disease [94]. In more recent studies using continuous glucose monitoring, patients with T2DM with greater MAGE had worse performance on the Montreal Cognitive Assessment, Trail-Making Test-B, and the Verbal Fluency Test [95]. In participants studied in the Atherosclerosis Risk in Communities (ARIC) study with a 20-year follow-up, levels of 1,5-AG, a biological marker of postprandial glucose elevations, were significantly associated with the risk of developing dementia [96]. While most studies above involved older adults with T2DM, amongst children with T1DM, children with larger fluctuations in glucose levels were also noted to have poorer performance on cognitive tests, particularly related to memory [97]. Other studies have shown that antecedent hypoglycemic episodes in children may adversely impact performance on cognitive tasks [98,99]. Finally, even amongst individuals without diabetes, long-term GV is associated with poorer performance in memory recall and verbal fluency tests in older subjects [100,101].

GV-driven CNS inflammation may also play an important role in acute CNS injury. In a study of 417 participants with acute coronary syndrome (ACS), short-term GV, measured by the mean amplitude of glycemic excursion, was an independent predictor of subsequent adverse cardiovascular and cerebrovascular events [102]. Similarly, amongst patients hospitalized for ACS in the intensive care unit, the standard deviation of inpatient blood glucose measurements was the strongest predictor for subsequent major cardiovascular events [103]. Numerous studies have also reported that higher GV is associated with poor outcomes following stroke [104,105,106] including decreased functional outcomes at discharge [104], cognitive impairments post stroke [105], and increased 3-month mortality [106]. Finally, in a study of individuals with traumatic brain injury, increased GV was associated with poorer long-term neurological outcomes [107]. While the mechanisms underlying these associations remain unclear, strong evidence indicates that both hyper- and hypoglycemia contribute to the inflammatory state that occurs post-CNS injury [108].

## 4. Strategies to Minimize GV

At the time of writing, there are nearly 70 clinical trials specifically investigating the impact of glycemic variability on clinical outcomes [109]. With the widespread use of newer technologies such as continuous glucose monitors to measure short-term GV, more publications have concluded that GV is an additional source of diabetic complications independent of glycemic control. These publications have questioned the use of HbA1C as the solo marker for diabetes treatment [110], evaluated the evidence from interventions to reduce GV [37], and proposed treatment plans that use GV as a component of diabetes management [111]. Both diabetes treatments and dietary choices have been found to reduce GV and the results of each will be explored here in turn.

### 4.1. Therapeutic Interventions to Minimize GV

Clinical studies have investigated the impact of both diabetes treatment factors as well as dietary factors that impact glycemic variability and several studies have indicated that reducing GV leads to a reduction in oxidative stress and a lower risk of developing diabetes-related complications. CGM use alone has been shown to improve glycemic control and minimizes glucose fluctuations [112,113]. In addition, studies on different modes of insulin administration have also shown that certain methods of administering insulin as treatment for T1DM achieve lower levels of GV. Continuous subcutaneous insulin infusion (CSII) has also been shown to achieve both better glycemic control and lower GV than multiple daily injection insulin administration [114]. Other treatment options available for diabetes control besides insulin have proven to be beneficial in reducing GV and inflammation including glucagon-like peptide-1 receptor antagonists (GLP-1 RA). One study found that use of the GLP1-RA, liraglutide, in conjunction with CGM in individuals with newly diagnosed T2DM improved not only glycemic control and glycemic variability, but also decreased oxidative stress markers [115]. Clinical trials with dipeptidyl peptidase 4 (DPP-4) inhibitors, which prevent the degradation of GLP1 have also been found to decrease MAGE as well as oxidative stress markers assessed by nitrotyrosine and inflammatory markers IL-6 and IL-18. Nitrotyrosine and IL-6 changes significantly correlated with changes in MAGE, but not in HbA1c [116]. Meglitinides, a class of antidiabetic agents which act on the KATP channels on pancreatic “β” -cells to induce insulin release, can reduce postprandial glucose excursions and GV. These medications have also been shown to reduce peripheral oxidative stress markers including a regression of carotid intima–media thickness and a reduction in markers of systemic vascular inflammation in individuals with T2DM [117,118].

### 4.2. Dietary Interventions to Minimize GV

A large number of dietary variables have been shown to affect glucose fluctuations, including diet composition and meal timing. Evidence has shown that the type and quantity of carbohydrates has the greatest influence on glycemic response [119]. Lin et al. correlated dietary components with glucose fluctuations in individuals with T1DM. They found that the group with a carbohydrate intake of <50% of their daily caloric intake had lower glucose fluctuations [120]. Other studies have shown that the supplementation of a high glycemic index meal with proteins such as whey protein reduces post meal glucoses by delaying gastric emptying, stimulating insulin and incretin secretion [121]. A study performed in subjects with obesity, in which two hypocaloric diets with similar macronutrient composition but different glycemic index were given, showed that adherence to a low glycemic index hypocaloric diet led to lower levels of GV as well as improved endothelial function when compared to high glycemic index foods [122]. Other studies have shown that the addition of certain ingredients can reduce glycemic response of foods. As an example, Henry et al. provided a low glycemic index (isomaltose) and high glycemic index (sucrose) diet to healthy men in a randomized, double-blind, controlled crossover design. The low glycemic index diet resulted in lower glycemic variability measured by CGM [123].

The order of food intake may also impact postprandial glucose levels. Shukla et al. showed that the temporal sequence of carbohydrate ingestion during a meal has a significant impact on postprandial glucose and insulin excursions [124]. Individuals were given identical meals of fixed portions of carbohydrates and protein but in a different order. Individuals who consumed the carbohydrates first were noted to have higher postprandial glucose excursions compared to those who had the protein component first [124]. In patients with type 2 diabetes, skipping breakfast has also been found to lead to higher GV. Subjects skipping breakfast had a lower fiber intake and higher carbohydrate-to-fiber ratio than those eating breakfast. While total calories per day were equal between the groups, omission of breakfast was associated with higher postprandial glucose responses after lunch and dinner, which may be mediated by lower glucagon-like peptide 1 and insulin secretion [125]. Finally, in a within-subject analysis study of women with gestational diabetes mellitus, high carbohydrate morning meals also had higher measures of MAGE but lower mean glucose compared to equicaloric diets that distributed carbohydrates throughout the day [126].

## 5. Conclusions

There is increasing evidence that glycemic variability is an independent driver of increased oxidative stress and inflammation, which can be particularly detrimental to CNS function. With greater use of technologies such as CGMs, which will allow for more rigorous quantifications of GV, future studies will be needed to define the exact relationships between GV, inflammation, and brain function.

## Figures and Tables

**Figure 1 nutrients-12-03906-f001:**
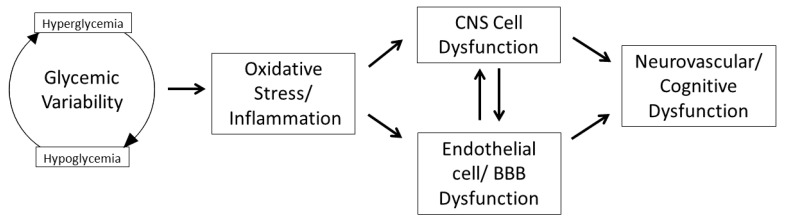
The impact of glycemic variability in the central nervous system.

**Table 1 nutrients-12-03906-t001:** Common terms used in the study of glycemic variability.

Abbreviation	Full Name	Description
Short-Term GV	Measures GV fluctuations in minute to hour increments	Typically measured using a CGM and reported with a variety of statistical measures
Long-Term GV	Measures GV on the scale of weeks to months	Typically reported as standard deviation of inter-visit HbA1C
HbA1C	GlycatedHemoglobin Assay	Reported as percentage of glycated hemoglobin
CGM	Continuous Glucose Monitor	A wearable medical device that regularly records blood sugar

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
