# Peer review of "Glycemic Variability and CNS Inflammation: Reviewing the Connection"

_nutrients, 2020, doi:10.3390/nu12123906_

Round 1
Reviewer 1 Report
This is an interesting review providing important insights on the connection between glycemic variability and neuroinflammation, drawing special attention to the impact that oxidative stress has on endothelial and vascular inflammation. The authors discussed the latest therapeutic implications as well as clinical and preclinical data, both in vivo and in vitro.
There are some majors concerns that need to be addressed to further strengthen this review article.
Major comments
1) Figure 1: The authors illustrated the established and potential relationships to glycemic variability. It is strongly recommended that the authors provide a more detailed scheme that would be helpful to enable the reader to see the detrimental role of glycemic variability on CNS system, also underlying its effects on brain endothelial function.
2) Recent studies from Kimura and Catargi [doi.org/10.1186/s12933-018-0761-5; doi.org/10.2337/dc18-2047] have also correlated an increased glycemic variability to the occurrence of adverse events in patients with diabetes undergoing percutaneous coronary revascularization. These works should be mentioned in the context of cardiovascular risk and brain health.
3) Section 3. More information about the connection between central nervous system resident cells and blood vessels should be provided, also to highlight the main topic of this review.
4) Line 217. The authors need to use a clear paragraph division to underline strategies from both dietary and diabetes treatment factors in reducing glucose fluctuations.
Author Response
We thank the reviewer for their very helpful comments and suggestions. Please see below for our revisions/changes.
Reviewer #1: Major comments
1) Figure 1: The authors illustrated the established and potential relationships to glycemic variability. It is strongly recommended that the authors provide a more detailed scheme that would be helpful to enable the reader to see the detrimental role of glycemic variability on CNS system, also underlying its effects on brain endothelial function.
We thank the reviewer for this very helpful suggestion, and we have revised our figure to now provide more mechanistic insight into the potential role of GV and its CNS consequences.
2) Recent studies from Kimura and Catargi [doi.org/10.1186/s12933-018-0761-5; doi.org/10.2337/dc18-2047] have also correlated an increased glycemic variability to the occurrence of adverse events in patients with diabetes undergoing percutaneous coronary revascularization. These works should be mentioned in the context of cardiovascular risk and brain health.
Thank you for bringing these additional studies to our attention. The paragraph has been revised in Section 3.3 (lines 218-226) to add them into the discussion on CVD risk and CNS outcomes.
3) Section 3. More information about the connection between central nervous system resident cells and blood vessels should be provided, also to highlight the main topic of this review.
Thank you for this suggestion and we have revised several sections in the review to highlight the impact of GV on 1) glucose transporters in the endothelial cells (Section 3 lines 138-142) and 2) the neurovascular unit (Section 3 lines 122-129).
4) Line 217. The authors need to use a clear paragraph division to underline strategies from both dietary and diabetes treatment factors in reducing glucose fluctuations.
We have now added sub-headings (Sections 4.1 and 4.2) to clearly delineate therapeutic and dietary factors. We have also added a sentence (line 237-238) to further make the separation.
Reviewer 2 Report
The review paper is well written and provides a concise overview of the topic. The authors have captured the key message bringing together a wealth of literature on the subject. The link to neuroinflammation is presented however, a detailed mechanism is not presented. That said, for a review of this type the relevant reference are included and so the reader is directed appropriately to further detail.
On occasion the authors have used excessively long sentences where the context can be difficult to comprehend and so it would be advisable to consider restructuring to ensure clarity eg lines 68-70 and 220-224.
Author Response
Reviewer #2:
We thank the reviewer for their very helpful comments and suggestions. Please see below for our revisions/changes.
The review paper is well written and provides a concise overview of the topic. The authors have captured the key message bringing together a wealth of literature on the subject. The link to neuroinflammation is presented however, a detailed mechanism is not presented. That said, for a review of this type the relevant reference are included and so the reader is directed appropriately to further detail. On occasion the authors have used excessively long sentences where the context can be difficult to comprehend and so it would be advisable to consider restructuring to ensure clarity eg lines 68-70 and 220-224.
We thank the reviewer for these very helpful comments. We have revised Figure 1 to provide more of a mechanistic framework to describe the impact of GV on the brain. In addition, we have revised the entire document for clarity, with particular attention to the sentences mentioned (lines 67-69 and lines 232-237).
Round 2
Reviewer 1 Report
I thank the authors for the diligently and thoroughly revised version of the article.